# A Novel Dual-Function Redox Modulator Relieves Oxidative Stress and Anti-Angiogenic Response in Placental Villus Explant Exposed to Hypoxia—Relevance for Preeclampsia Therapy

**DOI:** 10.3390/biology12091229

**Published:** 2023-09-12

**Authors:** Diana Pintye, Réka E. Sziva, Maxim Mastyugin, Brett C. Young, Sonako Jacas, Marianna Török, Saira Salahuddin, Prakash Jagtap, Garry J. Southan, Zsuzsanna K. Zsengellér

**Affiliations:** 1Department of Medicine, Beth Israel Lahey Health, Beth Israel Deaconess Medical Center, Harvard Medical School, Boston, MA 02115, USA; dpintye@bidmc.harvard.edu (D.P.); sjacas@bidmc.harvard.edu (S.J.); 2Department of Obstetrics and Gynecology, Semmelweis University, 1082 Budapest, Hungary; 3Department of Chemistry, University of Massachusetts, Boston, MA 02125, USA; maxim.mastyugin001@umb.edu (M.M.); marianna.torok@umb.edu (M.T.); 4Department of Obstetrics and Gynecology, Beth Israel Lahey Health, Beth Israel Deaconess Medical Center, Harvard Medical School, Boston, MA 02115, USA; bcyoung@bidmc.harvard.edu (B.C.Y.); ssalahud@bidmc.harvard.edu (S.S.); 5Akkadian Therapeutics, Stoneham, MA 02180, USA

**Keywords:** preeclampsia, nitric oxide-donor, redox modulator, SOD mimetic, placental villous explant, hypoxia

## Abstract

**Simple Summary:**

We intend to develop drug candidates for preeclampsia, a complication during pregnancy characterized by maternal high blood pressure, kidney dysfunction, severe cases of eclampsia, and premature birth. There is no cure or clear pathogenesis. This study confirmed that addressing the redox deficiencies and angiogenic imbalances in preeclampsia are valid targets for therapeutic intervention. This will have a significant impact on women who are pregnant and affected by preeclampsia, as well as on their babies.

**Abstract:**

Background: Preeclampsia (PE) is a severe, life-threatening complication during pregnancy (~5–7%), and no causative treatment is available. Early aberrant spiral artery remodeling is associated with placental stress and the release of oxygen radicals and other reactive oxygen species (ROS) in the placenta. This precedes the production of anti-angiogenic factors, which ultimately leads to endothelial and trophoblast damage and the key features of PE. We tested whether a novel dual-function redox modulator—AKT-1005—can effectively reduce placental oxidative stress and alleviate PE symptoms in vitro. Method: Isolated human villous explants were exposed to hypoxia and assessed to determine whether improving cell-redox function with AKT-1005 diminished ROS production, mitochondrial stress, production of the transcription factor HIF1A, and downstream anti-angiogenic responses (i.e., sFLT1, sEng production). MitoTEMPO was used as a reference antioxidant. Results: In our villous explant assays, pretreatment with AKT-1005 reduced mitochondrial-derived ROS production, reduced HIF-1A, sFLT1, and sEng protein expression, while increasing VEGF in hypoxia-exposed villous trophoblast cells, with better efficiency than MitoTEMPO. In addition, AKT-1005 improved mitochondrial electron chain enzyme activity in the stressed explant culture. Conclusions: The redox modulator AKT-1005 has the potential to intervene with oxidative stress and can be efficacious for PE therapy. Future studies are underway to assess the in vivo efficacy of HMP.

## 1. Introduction

Preeclampsia (PE) is a major disease of human pregnancy, marked by hypertension, proteinuria, and multisystem vasoconstriction appearing during the second or third trimester of gestation [1]. Incidence varies, subject to geographical region, time of year, nutrition, and race/ethnicity, and it affects 5–7% of all pregnancies [2].

The exact etiology and pathogenesis of PE are unclear, but there is growing evidence that placental ischemia, resulting from impaired placental remodeling and the subsequent abnormal secretion of anti-angiogenic factors, impairs mitochondrial function in the syncytiotrophoblast cells. The resulting production of reactive oxygen species (ROS) [3,4,5,6,7,8] and hypoxic stress have been demonstrated to stabilize hypoxia-inducible factor (HIF1A), which further induces transcription of anti-angiogenic factors, most notably soluble fms-like tyrosine kinase 1 (sFLT1) and soluble endoglin (sEng), in endothelial and trophoblast cells [9]. The ROS and reactive nitrogen species (RNS) originate from mitochondria in preeclamptic placentas, causing impaired mitochondrial function in both humans and animal models of PE [10,11,12].

The enhanced vasoconstrictor sensitivity, as well as elevations in sFLT1, precede clinical signs and symptoms of preeclampsia [13]. Indeed, overexpression of sFLT1 in pregnant mice induces angiotensin II sensitivity and hypertension by reducing endothelial nitric oxide (NO) flux, impairing phosphorylation of endothelial nitric oxide synthase (eNOS), and promoting oxidative stress in the vasculature [14]. Similarly, blockade of (nitric oxide synthase) NOS activity in pregnant mice with the inhibitor L-N^G^-Nitro arginine methyl ester (L-NAME) also induces angiotensin sensitivity and oxidative stress [14]. While nitro-vasodilators may correct NO deficiency, they do not correct superoxide radical (O_2_^•−^) excess, and as such, they do not resolve the free radical imbalance that causes profound vasoconstriction that can lead to local hypertension [15,16,17].

In this project, we tested AKT-1005, a novel dual-function NO-donor and redox modulator. AKT-1005 is an organic nitrate NO donor and a potent redox catalyst by virtue of its nitroxide functional group, which deactivates reactive oxygen species (ROS), but in particular superoxide (O_2_^•−^). These co-localized activities thereby avoid the reaction of the released NO with local O_2_^−^ that generates peroxynitrite (ONOO^−^). As a consequence, the flux of bioavailable NO from AKT-1005 is greater and the concentrations of deleterious ONOO^−^ much lower than for simple NO-donors.

As of today, there is no effective treatment available that addresses oxidative stress and the early contributors to the development of PE. To test the redox modulator effect of AKT-1005 in an ex vivo preeclampsia model, we used human villous explants and exposed them to hypoxia. Here we demonstrate that AKT-1005 protected mitochondrial function, attenuated oxidative stress and HIF1A stabilization, and diminished the subsequent anti-angiogenic response in hypoxia-exposed villous explants. These outcomes were compared with those of the antioxidant MitoTEMPO [12,18,19].

## 2. Materials and Methods

### 2.1. Materials

AKT-1005 (2,2,5,5-tetramethyl-1-pyrrolidinyloxy-3-yl)-methyl-nitrate was synthesized at Akkadian Therapeutics. MitoSOX™ Red was purchased from ThermoFisher Scientific (Waltham, MA, USA). Phosphate-Buffered Saline (PBS), DMEM/F12 medium, fetal bovine serum (FBS), trypsin, and penicillin-streptomycin were purchased from Gibco, Invitrogen (Carlsbad, CA, USA). MitoTEMPO was purchased from Sigma-Aldrich (Saint Louis, MO, USA).

### 2.2. Human Placental Villous Explant Cultures and Sample Collection

Placental samples were collected from patients with normotensive pregnancies and unlabored cesarian deliveries as described previously [20,21,22]. The exclusion criteria for the current study of controls were maternal infection, diabetes mellitus, multiple gestations, kidney disease, and fetal anomalies. The Institutional Review Board at Beth Israel Deaconess Medical Center approved the collection and use of discarded human placentas.

For explant culture, individual clusters of villous trees were dissected under a stereomicroscope and cultured in 1 mL of DMEM/F12 media containing 10% fetal bovine serum (FBS) and 1% antibiotic-antimycotic (Gibco, Carlsbad, CA, USA). Fresh 0.3 M stock solutions of AKT-1005 were dissolved in DMSO and further diluted in DMEM/F12 media, as recommended by the manufacturer (Akkadian Therapeutics, Stoneham, MA, USA). The culture medium was supplemented with 10, 50, or 100 μM of AKT-1005 or MitoTEMPO. MitoTEMPO stock solution was made at 10 mg/mL in sterile water and stored at −20 °C. The explants were maintained overnight at either 10% or 2% O_2_ with 5% CO_2_ at 37 °C.

### 2.3. Antioxidant Assay

The oxygen radical antioxidant capacity (ORAC) assay was carried out as described in our previous work [23]. Stock solutions of the reference antioxidants (ascorbic acid, Trolox, and MitoTEMPO) and the test compound, AKT-1005, were prepared by dissolving them in DMSO to a concentration of 10 or 50 mM, and then a serial dilution of the AKT-1005 stock solution was carried out by DMSO. After an additional 125X dilution with ethanol, 25 µL of each compound was added to a black flat-bottomed 96-well plate. A total of 150 µL aliquots of 0.08 µM fluorescein in a 75 mM, pH = 7.4 (37 °C) phosphate buffer were added to each well, followed by an incubation of the plate at 37 °C for 15 min before also adding 25 µL-s of a 153 mM 2,2′-azobis(2-amidinopropane) dihydrochloride (AAPH) solution prepared by dissolving AAPH in cold (4 °C) phosphate buffer. Each sample was prepared in triplicate. Control sets with AAPH and fluorescein with the appropriate amount of DMSO in ethanol were used to determine background absorbance. The positive control was the set with no AAPH (cold phosphate buffer was added instead) and only fluorescein. The negative control was the set with AAPH and no fluorescein (phosphate buffer was added instead). Fluorescence intensities were recorded every 2 min for 60 min using a SpectraMax i3x UV-Vis/fluorescence plate reader (Molecular Devices, San Jose, CA, USA) with the excitation and emission wavelengths at 485 nm and 520 nm, respectively. The data analysis was carried out according to the following equations:$$Net\;AUC=0.5+\sum_{0-29}\frac{f_{i}}{f_{0}}+\left(0.5\times \frac{f_{30}}{f_{0}}\right)$$
f_i_ is the fluorescence intensity at readings 0–29. f_0_ is the fluorescence at reading zero. f_30_ is the fluorescence at reading 30.
$$Percent\;Radical\;Scavenging=\frac{\left({Net\;AUC}_{t}-{Net\;AUC}_{c}\right)}{{(Net\;AUC}_{f\_max}-{Net\;AUC}_{c})}\times 100$$

The Net AUC_t_ is the area under the curve for the test sample. The Net AUC_c_ is the net area under the curve for the control sample with no compound (background absorbance). The Net AUC_f_max_ is the area under the curve for the maximum fluorescence sample where no AAPH was added (positive control).

### 2.4. Mitochondrial ROS Measurements

MitoSOX Red, a superoxide indicator, is a fluorogenic dye specifically targeted to mitochondria in live cells. Oxidation of the MitoSOX reagent by mitochondrial superoxide, but not by other ROS or RNS-generating systems, produces bright red fluorescence. Explants were cultured in 24-well plates (Nalgen Nunc International, Rochester, NY, USA) and incubated at 37 °C in a 10% CO_2_ humidified incubator overnight. The next day, cultures were subjected to hypoxia or normoxia treatment for 24 h, along with various concentrations of AKT-1005. After 24 h, the cultures were frozen in Tissue-Tek^®^ O.C.T. compound (Sakura, Torrance, CA) and cryosectioned and stored at −80 °C. The sections were incubated with MitoSOX™ Red (#M36008 ThermoFisher Scientific) fluorogenic dyes at 25 nM final concentrations, respectively, for 15 min. Sections were washed three times with PBS, and the specific red fluorescence was visualized and imaged using an Olympus microscope and cellSens|Imaging Software|Olympus LS [9].

### 2.5. HIF1A and CK8 Immunofluorescence

HIF1A protein expression was assessed by immunostaining. Briefly, explant cryosections were fixed in ice-cold acetone, permeabilization was achieved with 0.5% Triton X-100, slides were incubated with 1% BSA, and then followed by Alexa 488-anti-HIF1A antibody (anti-HIF-1 alpha antibody [EP1215Y] 1:75 dilution, ABCAM, Cambridge, MA, USA) overnight at 4 °C. The next day, it was incubated with Cytokeratin-8 (CK8 (a mouse anti-human antibody), which was visualized with Alexa Fluor^®^ 546—a conjugated horse anti-mouse secondary antibody. The nuclei were stained with 4′,6-diamidino-2-phenylindole (DAPI).

Morphometry was achieved with images acquired by fluorescence microscopy for MitoSOX™ Red and HIF1A and light microscopy for COX enzyme-histochemistry (original magnification of 40×). Four images were obtained and quantified per sample as replicates. Morphometric measurements were performed using Fiji software version 2 (National Institute of Health [NIH], Bethesda, MD, USA; https://imagej.net/software/fiji/ (accessed on 1 August 2023). To determine staining intensity, the mean optical density (OD) of MitoSOX™ Red fluorescence, HIF1A Green fluorescence, or 3,3′-diaminobenzidine staining was calculated for each image. To calculate OD per area, the mean intensity was divided by tissue area.

### 2.6. sFLT1, sEng, and VEGF Enzyme-Linked Immunosorbent Assay (ELISA)

Soluble FLT1 (sFLT1), soluble Endoglin (sEng), and vascular endothelial growth factor (VEGF) in culture medium were measured by ELISA using the human VEGF receptor 1 (VEGF R1), human Endoglin/CD105, and human VEGF Quantikine ELISA Kit (R&D Systems, Minneapolis, MN, USA) following the manufacturer’s instructions.

### 2.7. COX In Situ Enzyme Chemistry

Fresh-frozen explant preparations were cryosectioned and treated for COX enzyme chemistry as described previously [10,24,25]. Representative digital images of explant preparations (*n* = 3 to 4 per group) were acquired. Four images were obtained and quantified per sample as replicates. Morphometric measurements were performed using Fiji software, as described in Section 2.5.

### 2.8. Statistics

GraphPad Prism 10.0.2 statistical software (San Diego, CA, USA) was used for statistical analysis. First, the normality tests were carried out using Kolmogorov-Smirnov, Saphiro-Wilk, D’Agostino and Pearson, and Anderson-Darling tests in cases of normal distribution, followed by a parametric unpaired T-test or analysis of variance (ANOVA) with Tukey’s post hoc test. When the distribution was not normal, a non-parametric Mann-Whitney U test or Kruskal-Wallis test with Dunn’s post hoc test was performed. Data are presented either in mean ± standard error of mean (SEM) or in median [interquartile range/IQR]. Statistical significance: the *p*-value was less than 0.05 (*p* < 0.05).

## 3. Results

### 3.1. The Oxygen Radical Antioxidant Capacity (ORAC) Assay

The antioxidant capacity of AKT-1005 was assessed by the standard oxygen radical antioxidant capacity (ORAC) assay. It measures the signal intensity originated from a fluorescent probe, fluorescein, that is reduced in the presence of peroxyl radicals generated in situ by 2,2′-azobis(2-amidinopropane) dihydrochloride (APPH) [26,27,28]. Testing of AKT-1005 indicated strong radical scavenging in the ORAC assay, with an EC50/IC50 of 1.25 µM, calculated by the Sebaugh equation [29]. In Figure 1, we demonstrate AKT-1005′s superior antioxidant effect over ascorbic acid (AA) and Trolox (a vitamin E derivative) or the mitochondrial-targeted antioxidant MitoTEMPO [12,18,19,30]. These reference antioxidants exhibited a prominent reduction in percent radical scavenging as their concentration decreased to 12.5 μM, while AKT-1005 was still nearly 100% effective. AKT-1005 at 1.25 μM had approximately the same percent radical scavenging activity as Trolox at 12.5 μM. These trends were similar to our earlier observations with another nitroxide-type antioxidant molecule and a derivative of AKT-1005, 3-(hydroxymethyl)-1-oxy-2,2,5,5-tetramethylpyrrolidine (HMP). Our recent publication assessed HMP effects on stressed trophoblast cells and demonstrated favorable ROS scavenging activity under oxidative stress conditions [23].

We used MitoTEMPO as an antioxidant control because it has the same functional group as AKT-1005, the nitroxide. MitoTEMPO is a mitochondria-targeted superoxide dismutase (SOD) mimetic. It can convert superoxide molecules into hydrogen peroxide or oxygen and subsequently detoxify them to oxygen and water by catalase or glutathione peroxidase [19,31]. Additionally, we decided not to use Trolox as the reference antioxidant since it has not been proven to be a potent antioxidant in clinical trials [32,33,34].

We included ORAC data with combined AKT-1005 and HMP antioxidant activities in Appendix A to emphasize the superior effects of these compounds over reference antioxidants Trolox, ascorbic acid, and MitoTEMPO [23].

### 3.2. The Human Placental Explant Is an Efficient Culture System, and It Responds to Hypoxic Stimuli, as Occurs in Human Preeclampsia Disease

The expression of hypoxia-inducible factor-1α (HIF1A) was assessed in the villous explant culture subjected to hypoxia vs. normoxia. HIF1A is a transcription factor that has an important role in placental development. It has been reported that HIF1A is elevated in placentas from preeclamptic pregnancies [35,36]. Furthermore, HIF1A induces the expression of soluble vascular endothelial growth factor receptor-1 (sFLT-1), a key factor in preeclampsia [37,38]. As shown in the following immunofluorescent images, 2% O_2_ (hypoxic condition) increased HIF1A staining on the human villous explant (green fluorescence) (Figure 2). The syncytiotrophoblast cells were predominantly the site of HIF1A expression, as the double immunofluorescence with Cytokeratin-8 (CK8: red fluorescence, HIF1A + CK8 co-staining: yellow fluorescence) indicated. A smaller intensity of HIF1A staining is also present in the capillaries and in Hoffbauer cells. DAPI staining shows a nuclear counterstain with blue fluorescence.

### 3.3. AKT-1005 Dose-Dependently Reduced Mitochondria-Derived ROS Production in Hypoxia-Exposed Human Villous Explant

MitoSOX Red immunofluorescent staining indicates mitochondria-derived superoxide production. A 2% hypoxia exposure of the explant culture induced superoxide production (Figure 3a(D,D+), bright red immunofluorescence) compared to low levels in the normoxia group (Figure 3a(A,A+)) (*p* < 0.05). Both AKT-1005 (Figure 3a(E,E+)) and MitoTEMPO (Figure 3a(F,F+)) significantly decreased superoxide production (*p* < 0.05), but AKT-1005 resulted in significantly lower superoxide production than MitoTEMPO (*p* < 0.05) (Figure 3).

### 3.4. AKT-1005 Dose-Dependently Reduced HIF1A Transcription Factor Expression in Hypoxia-Exposed Villous Explant

HIF1A immunofluorescent staining (green color) indicates the increase in transcription factor expression in hypoxia-exposed explants (Figure 4a(D,D+)) compared to low-level expression in normoxia samples (Figure 4a(A,A+)). This increased HIF1A expression in response to hypoxia was significantly blunted in the presence of the antioxidants AKT-1005 (Figure 4a(E,E+)) and MitoTEMPO (Figure 4a(F,F+)) (*p* < 0.05), although AKT-1005 was more effective than MitoTEMPO (*p* < 0.05) (nearly to the level of normoxia) (Figure 4).

### 3.5. Pretreatment with AKT-1005 Dose-Dependently Reduced Anti-Angiogenic Expression and Increased Pro-Angiogenic Response in Hypoxia-Exposed Villous Explant

Hypoxic exposure of villous explants resulted in increased sFLT-1 and sEng expression in the culture supernatant, which was most likely the result of the upregulation of the HIF1A transcription factor in trophoblast cells. AKT-1005 pretreatment dose-dependently reduced sFLT-1 and sEng expression in the hypoxia-exposed explants (Figure 5). Moreover, AKT-1005 had the beneficial effect of increasing VEGF expression and the pro-angiogenic response in the hypoxia-exposed explants.

### 3.6. Compound AKT-1005 Pre-Treatment Improved Mitochondrial COX Expression in 2% O_2_-Exposed Human Villous Explant

Histochemical staining of COX enzymes indicates that hypoxia significantly decreased COX activity (mitochondrial electron transport chain enzyme—Complex 4) (*p* < 0.01) in the villous explant (Figure 6a(D,D+)) compared to the normoxic state (Figure 6a(A,A+)), which was restored by both AKT-1005 (Figure 6a(E,E+)) and the reference antioxidant MitoTEMPO (Figure 6a(F,F+)) (*p* < 0.01); moreover, AKT-1005 was more effective than MitoTEMPO (*p* < 0.01). (Figure 6a,b).

## 4. Discussion

In our human villous explant assays, pre-treatment with AKT-1005, a dual-function redox modulator compound, reduced mitochondrial-derived ROS production in hypoxia-exposed villous explants, demonstrating that a key factor in the development of PE, oxidative injury, can be attenuated by this compound. AKT-1005 also reduced downstream expression of the HIF-1A transcription factor. Consequently, AKT-1005 lowered sFLT1 protein expression in hypoxia-exposed villous explant cultures. AKT-1005 also improved the mitochondrial bioenergetics in the villous trophoblast cells, which is another promising outcome of the drug treatment. The present study used placental villous tissues from normal pregnancy placenta (stressed with hypoxia); however, we have to further study the efficacy of AKT-1005 in placental preparations from preeclamptic pregnancies to learn if the study drug can be beneficial in diseased tissue.

Reactive oxygen and nitrogen species (ROS and RNS) are thought to be involved in cell signaling, but in excess, they can cause cell death and the development and progression of several diseases, among them PE. Our human explant culture has proven useful to assess the efficacy of dual-function redox modulator therapy in alleviating oxidative stress due to hypoxia. Hypoxic exposure of the human villous explants in our model induced significant mitochondrial superoxide production. AKT-1005′s antioxidant properties have been documented to be superoxide dismutase mimetic; therefore, it is a valid compound to use in this model. AKT-1005′s superior efficacy over other antioxidants could be beneficial in in vivo models of PE.

The HIF1A transcription factor responds to low oxygen levels, initiating transcription of numerous genes during hypoxia. It is a heterodimeric molecule composed of two subunits: an oxygen-sensitive α subunit, which can rapidly degrade and inactivate during normoxia, and the constitutively active β subunit. HIF1A is highly expressed in the low-oxygen environment of the placenta, which occurs in early gestation. Therefore, its presence can be beneficial to placental development and function [36,39]. HIF1A has been shown to be overexpressed in placentas from preeclamptic women, along with sFLT-1 and sEng [40]. Adding to the importance of this transcription factor, if HIF1A is overexpressed in pregnant mice, they develop hypertension and kidney dysfunction [41]. Our studies showed that hypoxic exposure of villous explants induces expression of both HIF1A and the anti-angiogenic factor sFLT-1. This expression is prevented by pre-treatment with AKT-1005, consistent with the known induction of sFLT-1 expression by HIF-1A.

ROS production in the placenta during PE pregnancy has been shown by our group and other groups [6,8,10,42]. It can be advantageous in inducing signaling pathways, which can stimulate placental vascular development, invasion of trophoblasts into spiral arteries, differentiation, and transport in normal pregnancy. Nevertheless, if ROS production is sustained, it can cause damage to the trophoblast and endothelial cell mitochondria in the placenta, as we have demonstrated previously [10,21,43]. AKT-1005 clearly reduced MitoSOX Red, an indicator of mitochondrial-derived superoxide production, demonstrating less ROS production following treatment.

The AKT-1005 molecule contains both organic nitrate and nitroxide functional groups, through which it acts as both an NO donor and a potent redox catalyst, respectively. The nitroxide function rapidly deactivates superoxide (O_2_^•−^) and other reactive oxygen species (ROS). We recently reported that the AKT-1005 nitroxide analog HMP, which has no NO-donor capability, can reduce ROS generation in the stressed HTR-8/SVNeo cell line [23]. Moreover, we found that HMP downregulated HIF1A and the downstream target genes. It is conceivable that this anti-oxidant function is well-retained in the dual-function molecule AKT-1005 as well.

Nitric oxide reacts rapidly with superoxide to form peroxynitrite [44], and this occurs in vivo. Therefore, release of NO into a milieux containing superoxide (or other ROS) will consume NO and generate peroxynitrite (often seen as increased nitrotyrosine formation). Guzik et al. noted that in isolated tissues “NO donors and superoxide scavengers both reduced superoxide release, whereas only NO donors increased peroxynitrite formation”. AKT-1005, by removing superoxide in the same location as it releases NO, prevents the quenching of NO by superoxide, resulting in a greater molar yield of NO with the additional benefit of reduced formation of peroxynitrite. This is reflected in a reduction in nitrotyrosine staining in tissue from preeclamptic animals treated with AKT-1005 (publication pending).

It is well known that there is an increased expression of sFLT1 and sEng (anti-angiogenic response), along with decreased placental growth factor (PlGF) and vascular endothelial growth factor (VEGF) (pro-angiogenic response), which are all significant factors in PE pregnancies [38,45]. In our villous explant studies, we found similar abnormalities after hypoxic stress, i.e., an elevated anti-angiogenic response. We showed that AKT-1005 treatment reduced sFLT11 and sEng expression while the pro-angiogenic factor VEGF was increased, indicating that this dual-function drug can have a favorable effect on the expression of these angiogenic molecules. This is not entirely surprising since the expression of these genes is dependent on the activation of the HIF1A transcription factor, which was reduced after AKT-1005 treatment. The reduction in sFLT1 by AKT-1005 has relevance to the in vivo use of this compound as sFLT1 is a causative mediator of PE [46,47,48].

We have shown previously that the mitochondrial electron transport chain enzyme cytochrome C oxidase (COX) activity is diminished in the syncytiotrophoblast cells of the human placental villous tissue, implicating mitochondrial dysfunction as a potential contributing factor to the pathogenesis of PE [10]. In this study, we similarly found a reduction in mitochondrial COX function in our hypoxia-exposed villous explant, mainly in syncytiotrophoblast cells. In contrast, after AKT-1005 treatment, COX activity improved, revealing the beneficial effect of the drug treatment on mitochondrial bioenergetics.

## 5. Conclusions

In conclusion, the human placental explant is an efficient culture system, and it responds to hypoxic stimuli, as occurs in the human preeclampsia disease. Pretreatment with AKT-1005 dose-dependently reduced mitochondrial-derived ROS production as well as HIF1A transcription factor expression in hypoxia-exposed villous explants. Downregulation of sFLT1 by AKT-1005 is a significant factor in preventing downstream pathways and potentially the development of preeclampsia. Finally, AKT-1005 improved the mitochondrial activity in the trophoblast cells, which is another promising characteristic of this compound. Future studies are planned to investigate the in vivo use of AKT-1005 for preeclampsia therapy.

## Figures and Tables

**Figure 1 biology-12-01229-f001:**
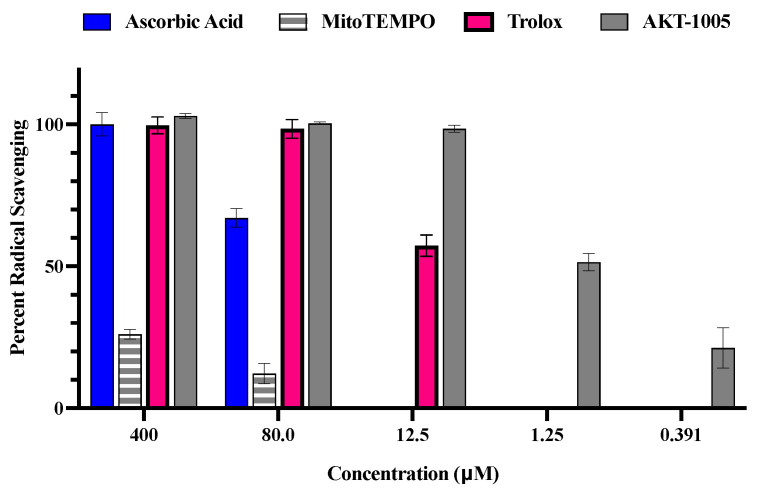
**ORAC assay: AKT-1005 antioxidant activity compared to ascorbic acid, MitoTEMPO, and Trolox at different concentrations**. (*n* = 3 per group).

**Figure 2 biology-12-01229-f002:**
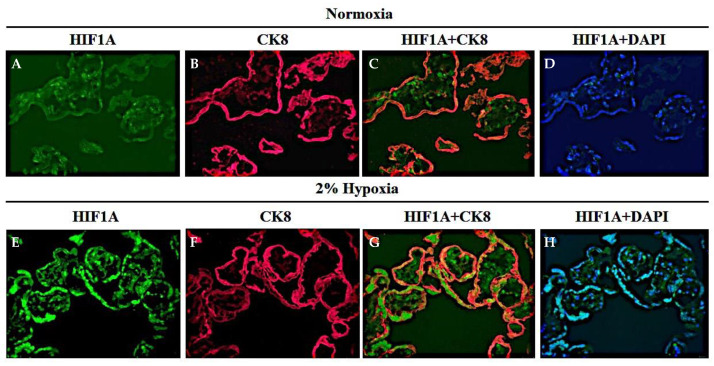
**The human placental explant is an efficient culture system, and it responds to hypoxic stimuli, as occurs in human preeclampsia disease.** Human villous explant preparations were subjected to normoxia (10% O_2_) (**A**–**D**) or hypoxia (2% O_2_) (**E**–**H**) for 24 h. HIF1A expression was increased after hypoxic exposure (**E**) compared to a normoxic state (**A**). The explant preparations were co-stained with CK8 (trophoblast epithelial marker) (**C**,**G**), indicating that the majority of HIF1A localizes to syncytiotrophoblast cells. Counterstaining: DAPI (**D**,**H**), blue nuclear staining).

**Figure 3 biology-12-01229-f003:**
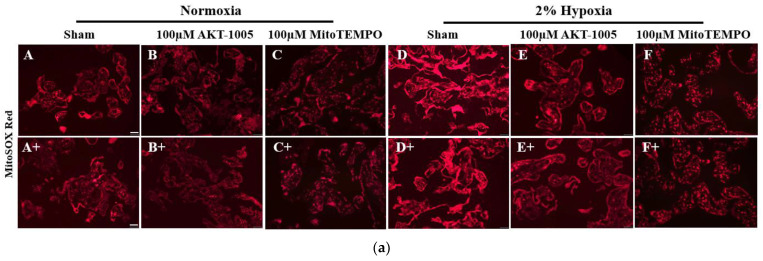
**Compound AKT-1005 pre-treatment reduced mitochondrial-derived ROS production in 2% O_2_-exposed human villous explants.** (**a**) Immunofluorescent staining of MitoSOX Red (A–F and A+–F+, superoxide) in human villous explants treated with 100 μM AKT-1005 and 100 μM MitoTEMPO for 30 min, then exposed to 24 h of 2% hypoxia. A bright red color indicates the intensity of MitoSOX Red immunofluorescent staining and correlates with the level of superoxide production. Scale bar: 200 µm. (**b**) Quantitation of MitoSOX Red immunofluorescence in trophoblasts: the mean optical density of whole cell surface area was calculated in four high-power fields per sample (*n* = 4 per group). Mann-Whitney U test, median [IQR]. Abbreviation: MT = MitoTEMPO. Normoxia vs. Hypoxia: *: *p* < 0.05; Hypoxia vs. H+100 μM AKT-1005: *: *p* < 0.05; Hypoxia vs. H+100 µM MT: *: *p* < 0.05; and H+100 µM AKT-1005 vs. H+100 µM MT: *: *p* < 0.05.

**Figure 4 biology-12-01229-f004:**
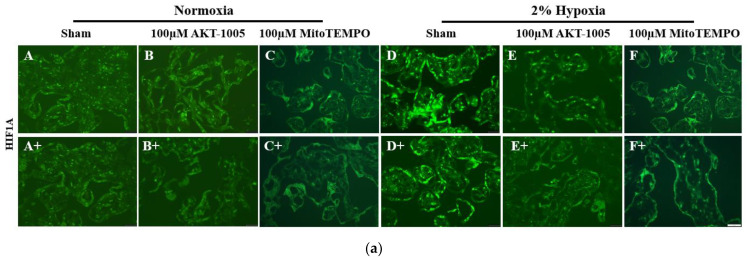
**Compound AKT-1005 pre-treatment reduced HIF1A expression in 2% O_2_-exposed human villous explants.** (**a**) Immunofluorescent staining of HIF1A transcription factor expression (A–F and A+–F+) in human villous explants treated with 100 μM AKT-1005 and 100 μM MitoTEMPO for 30 min, then exposed to 24 h of 2% hypoxia. A bright green color indicates HIF1A expression. Scale bar: 200 µm. (**b**) Quantitation of HIF1A immunofluorescence in trophoblasts: the mean optical density of whole cell surface area was calculated in four high-power fields per sample (*n* = 4 per group). Mann-Whitney U test, median [IQR]. Abbreviation: MT = MitoTEMPO. Normoxia vs. Hypoxia: *: *p* < 0.05; Hypoxia vs. H+100 µM AKT-1005: *: *p* < 0.05; Hypoxia vs. H+100 µM MT: *: *p* < 0.05; and H+100 µM AKT-1005 vs. H+100 µM MT: *: *p* < 0.05.

**Figure 5 biology-12-01229-f005:**
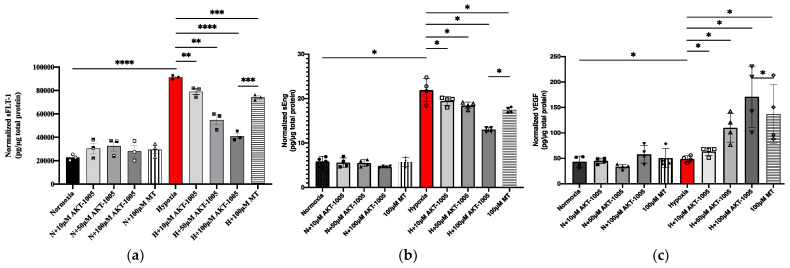
**AKT-1005 dose-dependently reduced (a) sFLT-1 protein and (b) sEng expression (anti-angiogenic response) and increased (c) VEGF (pro-angiogenic response) in human villous explant culture during hypoxic exposure.** Normalized sFLT-1/sEng/VEGF data according to different treatment groups). Abbreviations: N = normoxia; H = hypoxia; MT = MitoTEMPO. Figure (**a**): *n* = 3 per group, Shapiro-Wilk normality test. Unpaired T-tests. mean ± SEM. Figure (**b**,**c**): *n* = 4 per group, Mann-Whithey-U test, median [IQR]. Mean ± SEM. ****: *p* < 0.0001; ***: *p* < 0.001; **: *p* < 0.01; *: *p* < 0.05.

**Figure 6 biology-12-01229-f006:**
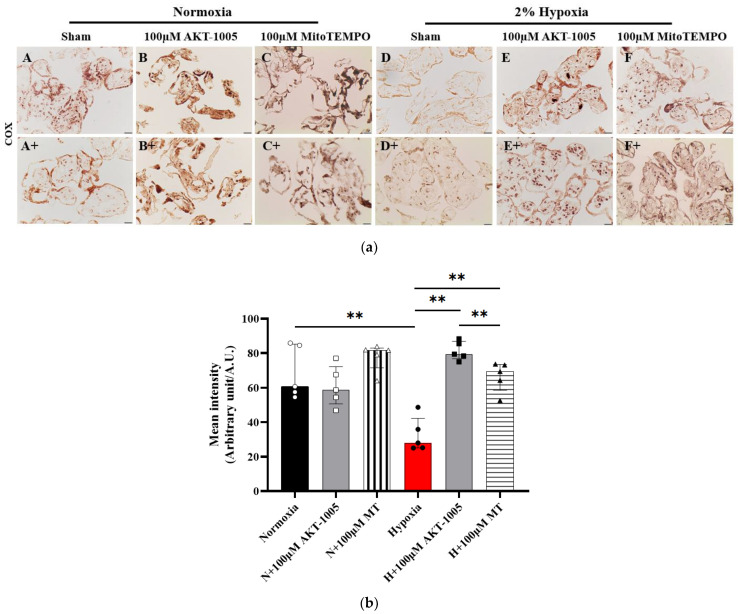
**Compound AKT-1005 antioxidant pre-treatment improved mitochondrial COX activity in 2% O_2_-exposed human villous explants.** (a) Representative COX enzyme histochemistry (A–F and A+–F+) in different groups of villous explants treated with 100 μM AKT-1005 and 100 μM MitoTEMPO for 30 min, then exposed to 2% hypoxia for 24 h. Brown staining represents COX activity. Scale bar: 200 µm. (b) COX enzyme activity in trophoblasts was quantitated by calculating the mean optical density of the DAB products in four 40× power fields per sample *(n* = 5 per group). Mann-Whitney U test, median [IQR]. Abbreviation: MT = MitoTEMPO. Normoxia vs. Hypoxia: **: *p* < 0.01; Hypoxia vs. H+100 µM AKT-1005: **: *p* < 0.01; Hypoxia vs. H+100 µM MT: **: *p* < 0.01; and H+100 µM AKT-1005 vs. H+100 µM MT: **: *p* < 0.01.

## Data Availability

All data can be accessed with the approval of Akkadian Therapeutics.

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
