# Peer review of "A Novel Dual-Function Redox Modulator Relieves Oxidative Stress and Anti-Angiogenic Response in Placental Villus Explant Exposed to Hypoxia—Relevance for Preeclampsia Therapy"

_biology, 2023, doi:10.3390/biology12091229_

Round 1

Reviewer 1 Report

This is a great paper!

Only a few comments divided into syntax and grammar section and then content

Syntax and grammar

Line 208: Can consider changing to: to emphasize the superior (can delete “on”)

Line 330: I am not sure if regenerated is the word you wanted? Recapitulated?

Content

Line 128-129: Consider providing an “in brief” overview of how the ORAC assay is done for the convenience of the reader, even though it was described previously

Lin3 132-140: It would be great if the authors could provide some insight into how MitoSOX assay measures mitochondrial ROS (a bit of insight into the chemistry)

Figure 3: The MitoTEMPO treated images to me look like the syncytial superoxides were largely abrogated and the positivity is in the villous stroma whereas the AKT treated images look like the post treated superoxides are present in syncytium and stoma. What do the authors think about this?

Figure 5: AKT seems to have increased VEGF levels in explants above the levels seen in normoxia (P value fort this comparison not shown). If that is true di the authors have a theory for why

Also looking at part c the hypoxia and normoxia levels look super close but there is a * indicating significance; can the authors explain this in more detail

Reviewer 2 Report

In the present study, the authors compare the antioxidant effects of a novel dual function NO-donor and redox modulator (AKT-1005) with those observed with another antioxidant (mito-Tempo), in human villous explant exposed to hypoxia. The authors conclude that AKT-1005 has a greater effect reducing mitochondrial-derived ROS production, HIF-1A expression and sFLT-1 levels  than MitoTempo in hypoxia exposed villous explants. They also conclude that dowregulation of sFLT1 by AKT-1005 is as significant factor preventing the dowstream pathways and potentially the development of preeclampsia. It is an interesting study, however there are some issues that need to be clarified.

1.-The authors state in the introduction that “flux of bioavailable NO from AKT-1005 is greater and the concentrations of deleterious ONOO- much lower than for simple NO-donors”.  However, there is no reference in the text to support this statement. The authors should add the reference or explain on what basis they make this assertion.

2.- Taking into account that, according to Figure 1, mitoTEMPO is the antioxidant with the lowest radical scavenging capacity. Why do the authors compare the antioxidant capacity of the AKT-1005 with that of mitoTempo and not with Trolox?.

3.- The data of immunofluorescence quantification are not shown in section 3.2 of the results. They should be included.

4.- Authors should specify whether they have used a one-way or two-way ANOVA. A two-way ANOVA would be more correct to compare the different experimental groups.

5.- The authors use as a model human villous explants from women with normal pregnancies, collected at the end of the third trimester of gestation. This implies, theoretically,  that these placentas have been subjected to adequate oxygen conditions throughout gestation. In contrast, a preeclamptic placenta would have been subjected to hypoxia from early gestation and these conditions could alter the characteristics and function of trophoblasts and endothelial cells in a way that the response to the drug might be different. Furthermore, the fact that the drug has an antioxidant effect on the placentas of normotensive pregnancies at the end of the third trimester does not mean that it can have the same effect in early gestation. The results obtained should be interpreted with caution and the authors should discuss these aspects.

6.-In the discussion, lines 314-316, the authors state that “....indicating that the key factor in the development of PE...”. Oxidative stress in one of the factors contributing to the development of the disaease but but it has not been proven to be the key factor.

7.-In the discussion, lines 342-346 the authors state that “In line with these results, the subsequent measurement of anti-angio-genic factor sFLT-1 expression has corroborated that the AKT-1005 effect on HIF1A has downstream effect on reducing sFLT-1 expression”. The results of the study do not demonstrate that the decrease in HIF1A mediates the reduction in sFLT-1 levels observed with AKT- 1005.  Both AKT-1005 effects are associated but the authors cannot conclude from the results that the decrease in sFLT-1 is caused by the action of HFL1A.
